# Evaluation of Colonoscopy and Sigmoidoscopy Utilization for Colorectal Cancer Screening in Georgia, USA

Benjamin E. Ansa [1,*], Zachary Hoffman [2], Nicollette Lewis [3], Biplab Datta [1], K. Monirul Islam [1] and J. Aaron Johnson [1]

1   Institute of Public and Preventive Health, Augusta University, Augusta, GA 30912, USA
2   Master of Science in Experimental Psychology Program, Augusta University, Augusta, GA 30912, USA
3   Medical College of Georgia, Augusta University, Augusta, GA 30912, USA
*   Correspondence: bansa@augusta.edu; Tel.: +1-706-721-6141

**Abstract:** Colorectal cancer (CRC) is the third most prevalent cancer, and the second most common cancer-related cause of death in the United States (USA). Timely screening reduces both CRC incidence and mortality. Understanding population behaviors and factors that influence CRC screening is important for directing interventions targeted at reducing CRC rates. The 1997–2018 Behavioral Risk Factor Surveillance System (BRFSS) data were analyzed for trends in colonoscopy and sigmoidoscopy utilization for CRC screening among adults in Georgia, USA. Overall, in Georgia, there has been an increase in the prevalence of colonoscopy and sigmoidoscopy utilization from 48.1% in 1997 to 71.2% in 2018 (AAPC = 2.30, $p < 0.001$). Compared nationally, this increase was less pronounced (from 41.0% in 1997 to 73.7% in 2018 (AAPC = 2.90, $p < 0.001$) overall for USA). Logistic regression analysis of the 2018 BRFSS data, adjusting for sociodemographic factors, shows that sex (female vs. male [aOR = 1.20, C.I. = 1.05, 1.38]); marital status (couple vs. single [aOR = 1.20, C.I. = 1.04, 1.39]); healthcare coverage (yes vs. no [aOR = 3.86, C.I. = 3.05, 4.88]); age (60–69 years [aOR = 2.38, C.I. = 2.02, 2.80], 70–79 [aOR = 2.88, C.I. = 2.38, 3.48] vs. 50–59 years); education (high school [aOR = 1.32, C.I. = 1.05, 1.65], some post high school [aOR= 1.63, C.I. = 1.29, 2.06], college graduate [aOR = 2.08, C.I. = 1.64, 2.63] vs. less than high school); and income ($25,000–$49,999 [aOR = 1.24, C.I. = 1.01, 1.51], $50,000+ [aOR = 1.56, C.I. = 1.27, 1.91] vs. <$25,000) were all significantly associated with colonoscopy and sigmoidoscopy utilization. In Georgia, a significant increase over time in colonoscopy and sigmoidoscopy utilization for CRC screening was observed pertaining to the associated sociodemographic factors. The findings from this study may help guide tailored programs for promoting screening among underserved populations.

**Keywords:** colorectal cancer; screening; colonoscopy; sigmoidoscopy; Georgia; USA

## 1. Introduction

Colorectal cancer (CRC) is a major public health burden in the United States (USA) [1], with an estimated 151,030 new cases and 52,580 deaths from the disease expected to occur in 2022 [2]. Timely screening facilitates the identification and removal of precancerous lesions and prevents the development of CRC [3,4]. Screening also reduces the incidence and mortality from CRC [5,6]. In 2021, the American College of Gastroenterology (ACG) recommended CRC screening in average-risk individuals between the ages of 45 and 75 [7,8]. There are several CRC screening modalities currently available including blood stool tests, sigmoidoscopy, and colonoscopy [7–9].

Fecal immunochemical testing (FIT) detects CRC with 91% sensitivity and 90% specificity, gFOBT has a sensitivity of 50–75%, and flexible sigmoidoscopy provides direct visualization of the distal colon and has a 90–100% sensitivity for CRC in the distal colon [8]. Colonoscopy has a sensitivity of 73–89% and specificity of 93% [10,11]. The FIT and fecal

occult blood test (FOBT) reduce mortality from CRC by 40% and 15–33% respectively, compared to 13–50% for sigmoidoscopy and 60–75% for colonoscopy [12–15].

Despite the proven advantages of screening, rates for CRC screening remain low in the USA. The 2018 National Health Interview Survey (NHIS) data reveals that only 65.2% of eligible adults have met the guidelines for CRC screening in the USA. Several factors associated with screening rates include age, race/ethnicity, education, insurance coverage, and geographic location [16,17]. The Healthy People 2030 target is to increase to 74.4% the adults aged 50 to 75 who have received a CRC screening test based on the most recent guidelines [17,18]. The rates and preferences for the different methods of CRC screening have been assessed among populations and geographic locations [19–23]. However, very few published data exist with regards to CRC screening behaviors in the state of Georgia [24,25]. The aims of the current study were to assess the prevalence and trends of colonoscopy and sigmoidoscopy utilization for CRC screening in Georgia, USA, and to determine the associated sociodemographic factors.

## 2. Materials and Methods

### 2.1. Data Source and Study Participants

The 1997 to 2018 datasets from the Behavioral Risk Factor Surveillance System (BRFSS) were analyzed for this study. The BRFSS is a state-based survey of the noninstitutionalized U.S. adult civilian population [26,27]. It conducts more than 400,000 adult interviews each year through random-digit-dialed telephone survey and collects data on residents in all 50 states, the District of Columbia, and three USA territories regarding their health-related risk behaviors, chronic health conditions, and use of preventive services [26]. The state of Georgia has contributed to the system since it was established in 1984 [28]. The standards set by the American Association of Public Opinion Research (AAPOR) Response Rate Formula 4 [29] are utilized for calculating BRFSS response rates. In 2018, the survey response rates for all states, territories and Washington, D.C. ranged from 38.8% to 67.2% with a median of 49.9% [30]. For the present study, the 2018 data for Georgia was analyzed, and the combined response rate for cell phone and landline was 43.6 [30].

The study participants were adults 50 years and older from Georgia who responded "Yes" or "No" to the question of if they have ever had a colonoscopy or sigmoidoscopy for CRC screening.

### 2.2. Measures

The trends in colonoscopy and sigmoidoscopy utilization based on sociodemographic variables from 1997 through 2018 were assessed. The prevalence and odds of colonoscopy and sigmoidoscopy utilization were calculated from the 2018 dataset only. The predictors were the following sociodemographic variables: sex, race, education, annual income, marital status (single relationship (divorced, widowed, separated, never married), couple relationship (married or a member of an unmarried couple)), healthcare coverage, and age. The outcome variables were (1) ever had colonoscopy or sigmoidoscopy (yes/no), (2) colonoscopy in the past ten years (yes/no), and (3) sigmoidoscopy in the past five years (yes/no).

### 2.3. Statistical Analysis

The yearly percentages of respondents from 1997 to 2018 who have ever had a colonoscopy or sigmoidoscopy were calculated for Georgia and the USA from the on-line BRFSS Prevalence Data & Data Analysis Tools [31]. The average annual percent change (AAPC) was calculated for changes in percentages of respondents utilizing colonoscopy or sigmoidoscopy over time. The Joinpoint Regression Program Version 4.5.0 (NCI, Rockville, MD, USA) [32] was used for calculating AAPC.

Descriptive statistics of respondents related to colonoscopy or sigmoidoscopy utilization were generated for 2018 using frequencies and proportions. Crosstabs were done to calculate weighted percentages of respondents who reported having had a colonoscopy or

sigmoidoscopy in 2018. Data were weighted to adjust for non-coverage, non-response, and for generalization of results [33].

The association between colonoscopy and sigmoidoscopy utilization and respondents' characteristics were determined from binary logistic regression analyses of the 2018 data. Adjusted odds ratios and related 95% confidence intervals were calculated. Data were adjusted for sex, age, race, marital status, education, income, and healthcare coverage. The significance level was set at $p < 0.05$, and all tests were two-sided. Unweighted counts, weighted percentages, and logistic regression analyses were performed using the IBM SPSS version 28 (IBM Corp., Armonk, NY, USA) [34].

### 2.4. Ethical Considerations

Publicly accessible BRFSS data do not contain personally identifiable information, therefore Institutional Review Board (IRB) approval was not necessary for this study. The process of data collection and release are governed by appropriate rules, regulations, and legislative authorizations [35].

## 3. Results

### 3.1. Average Annual Percent Change (AAPC) in Colonoscopy or Sigmoidoscopy Utilization

There was an overall increase in colonoscopy or sigmoidoscopy utilization for CRC screening between 1997 and 2018 for the state of Georgia from 48.1% to 71.2% (AAPC = 2.3, $p < 0.001$) and nationwide from 41.0% to 73.7% (AAPC = 2.9, $p < 0.001$) (Figure 1). Although not displayed in this report, inflexion points were observed in the data. For Georgia, there was a significant rise in colonoscopy or sigmoidoscopy utilization from 1997 to 2012 (AAPC = 2.8, $p < 0.001$), followed by a non-significant increase from 2012 to 2018 (AAPC = 0.4, $p = 0.8$). The colonoscopy or sigmoidoscopy utilization rate increased sharply nationwide from 1997 to 2008 (AAPC = 4.0, $p < 0.001$), and then steadily from 2008 to 2018 (AAPC = 1.8, $p < 0.001$).

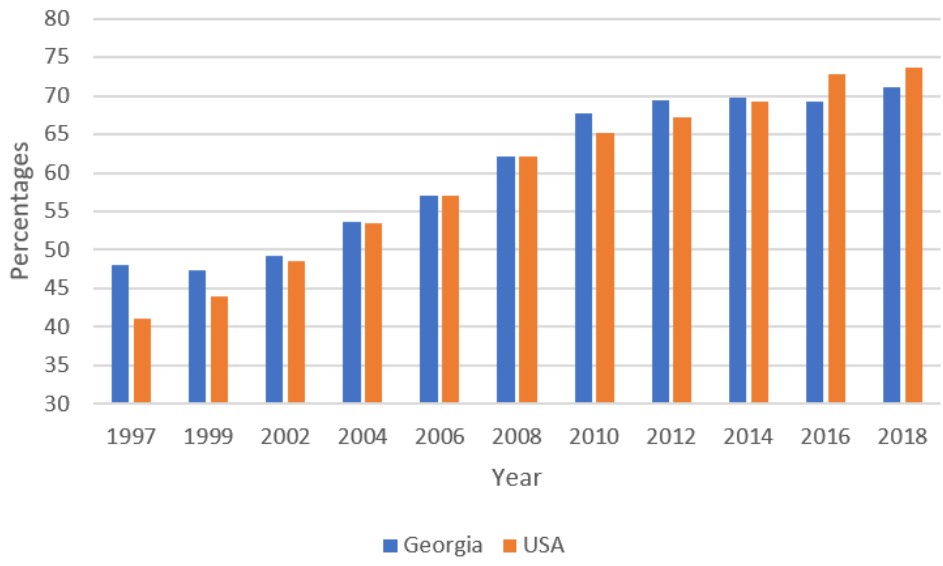

**Figure 1.** Adults aged 50+ who have ever had a colonoscopy or sigmoidoscopy: 1997–2018 BRFSS data.

From 1997 to 2018, a significant increase in colonoscopy or sigmoidoscopy utilization was observed among all sociodemographic categories (Table 1). However, the increase was higher for respondents who were Black (AAPC = 3.1, $p < 0.001$), female (AAPC = 2.6, $p < 0.001$), with a high school education (AAPC = 2.5, $p < 0.001$), and earning $35,000–$49,999 annually (AAPC = 3.3, $p < 0.001$).

**Table 1.** Crude prevalence of adults aged 50+ who have ever had a colonoscopy or sigmoidoscopy in Georgia: 1997–2018 BRFSS Data.

| Variable | 1997 (%) | 1999 (%) | 2002 (%) | 2004 (%) | 2006 (%) | 2008 (%) | 2010 (%) | 2012 (%) | 2014 (%) | 2016 (%) | 2018 (%) | AAPC * |
|---|---|---|---|---|---|---|---|---|---|---|---|---|
| Overall (Georgia) | 48.1 | 47.4 | 49.2 | 53.7 | 57.0 | 62.2 | 67.7 | 69.4 | 69.8 | 69.2 | 71.2 | 2.3 |
| **Sex** | | | | | | | | | | | | |
| Male | 51.6 | 51.3 | 47.4 | 54.4 | 55.4 | 61 | 67 | 69.4 | 67.6 | 69.9 | 69.7 | 2.0 |
| Female | 45.2 | 44.3 | 50.6 | 53.2 | 58.4 | 63.2 | 68.3 | 69.5 | 71.7 | 68.8 | 72.2 | 2.6 |
| **Age (years)** | | | | | | | | | | | | |
| 50–59 | 40.9 | 38.3 | 41.4 | 45.1 | 47.1 | 50.9 | 57.5 | 58.8 | 57.8 | 56.6 | 53.8 | 2.1 |
| 60–64 | - | 55.1 | 49.8 | 59.3 | 61.8 | 67.4 | 75.6 | 73.1 | 73.1 | 71.4 | 73.3 | 2.0 |
| 65+ | 53.6 | 53.9 | 57.9 | 62.1 | 67.2 | 73.9 | 76.3 | 79.1 | 80.4 | 75.3 | 78.5 | 2.1 |
| **Race** | | | | | | | | | | | | |
| White | 50.5 | 49.3 | 51.4 | 55.2 | 59.1 | 64.1 | 69.9 | 71.2 | 72.3 | 72.4 | 74.2 | 2.3 |
| Black | 41.0 | 37.7 | 41.4 | 52.3 | 52.4 | 56.5 | 61.6 | 65.2 | 69.7 | 64.1 | 69.4 | 3.1 |
| **Education** | | | | | | | | | | | | |
| <High School | 34.7 | 43.1 | 43.3 | 45.2 | 42.9 | 52 | 53.9 | 58.5 | 52.7 | 53.1 | 55.4 | 2.0 |
| High School | 48 | 38.6 | 44.2 | 51 | 54.4 | 57.1 | 62.3 | 66.4 | 68.3 | 65.4 | 66.0 | 2.5 |
| Some Post High School | 57.6 | 50.3 | 48.9 | 53 | 57.8 | 63.5 | 70.1 | 73.8 | 74.6 | 72.1 | 72.0 | 2.0 |
| College Graduate | 54.8 | 61.7 | 59.9 | 62.4 | 64.5 | 70.7 | 76 | 76.8 | 78.2 | 76.7 | 78.6 | 1.8 |
| **Income** | | | | | | | | | | | | |
| <$15,000 | 37.2 | 37.5 | 43.2 | 45.8 | 49.8 | 54.6 | 52.1 | 55.8 | 56.8 | 56.3 | 57.7 | 2.3 |
| $15,000–$24,999 | 50.3 | - | 48.7 | 51.8 | 52.1 | 55.8 | 61.5 | 60.7 | 61.4 | 65.0 | 63.6 | 1.5 |
| $25,000–$34,999 | - | - | 51.4 | 56.9 | 56.7 | 60.8 | 61.5 | 69.9 | 74.1 | 67.2 | 68.0 | 1.9 |
| $35,000–$49,999 | - | - | 44.5 | 47.3 | 58.7 | 62.6 | 69.1 | 70.2 | 75.2 | 69.8 | 76.4 | 3.3 |
| $50,000+ | 59.9 | 52.9 | 56.5 | 57.5 | 60.3 | 68.2 | 74.1 | 77.7 | 75.7 | 77.6 | 78.4 | 2.0 |
| Nationwide Overall (USA) | 41.0 | 43.9 | 48.6 | 53.5 | 57.1 | 62.2 | 65.2 | 67.3 | 69.3 | 72.9 | 73.7 | 2.9 |

* AAPC: Average annual percent change. AAPCs were statistically significant for all values at *p* = 0.05.

### 3.2. Characteristics of Study Population

There were three sets of survey respondents included in the analysis of the 2018 BRFSS data (Table 2). The first set (N = 5211) responded "yes" or "no" to the question if they have ever had a colonoscopy or sigmoidoscopy. The second set (N = 3947) responded "yes" or "no" to if they have had a colonoscopy in the past ten years, and the third set responded "yes" or "no" to if they have had a sigmoidoscopy within the past five years. For the three sets of respondents, the majority were White (≥65%), female (>56%), less than 70 years (>59%), in a couple relationship (>50%), and had healthcare coverage (>83%). Most of them were college graduates (>34%), earning $50,000 or more annually (>36%).

### 3.3. Prevalence of Colonoscopy and Sigmoidoscopy Utilization for Colorectal Cancer Screening

Sociodemographic differences in colonoscopy and sigmoidoscopy utilization for CRC screening were assessed from the 2018 BRFSS data. The unweighted frequencies and weighted percentages for respondents who answered "yes" to the colonoscopy and sigmoidoscopy survey are displayed in Table 3. Most of the respondents (62.9%) reported having had a colonoscopy in the past ten years, and comparatively much fewer respondents (3.5%) reported having had a sigmoidoscopy in the past five years. A combined 68.3% of respondents have ever had a sigmoidoscopy or colonoscopy. The prevalence of colonoscopy and sigmoidoscopy utilization was higher for older (70–79 years) female respondents with healthcare coverage. However, variations in the use of the two screening modalities were observed among sociodemographic categories of race, education, income, and marital status. The prevalence of colonoscopy utilization was highest for respondents who were Asian (69.4%), college graduates (72.1%), earning $50,000+ annually (70.3%), and in a couple relationship (67.1%). In contrast, the prevalence of sigmoidoscopy utilization

was highest for respondents who were Black (4.8%), with a less than high school education (5.0%), earning less than $25,000 annually (4.1%) and in a single relationship (3.8%).

**Table 2.** Characteristics of respondents to the colonoscopy and sigmoidoscopy survey in Georgia: BRFSS 2018 Data.

| Variable | Respondents to the Question "Have You Ever Had a Colonoscopy or Sigmoidoscopy?" | Respondents to the Question "Have You Had a Colonoscopy in the Past 10 Years?" | Respondents to the Question "Have You Had a Sigmoidoscopy within the Past 5 Years?" |
|---|---|---|---|
| Overall | N = 5211 (%) | N = 3947 (%) | N = 1859 (%) |
| **Sex** | | | |
| Male | 2114 (40.6) | 1671(42.3) | 815 (43.8) |
| Female | 3095 (59.4) | 2275 (57.6) | 1044 (56.2) |
| Don't Know/Refused | 2 | 1 | - |
| **Age (years)** | | | |
| 50–59 | 1486 (28.5) | 1430 (36.2) | 806 (43.4) |
| 60–69 | 1744 (33.5) | 1670 (42.3) | 743 (40.0) |
| 70–79 | 1300 (24.9) | 847(21.5) | 310 (16.7) |
| 80+ | 545 (10.5) | - | - |
| Don't Know/Refused | 136 (2.6) | - | - |
| **Race** | | | |
| White NH | 3489 (67.0) | 2601 (65.9) | 1208 (65.0) |
| Black NH | 1204 (23.1) | 953 (24.1) | 411 (22.1) |
| Hispanic | 189 (3.6) | 166 (4.2) | 116 (6.2) |
| American Indian/Alaskan Native | 50 (1.0) | 39 (1.0) | 20 (1.1) |
| Asian | 36 (0.7) | 29 (0.7) | 16 (0.9) |
| Native Hawaiian/Pacific Islander | 5 (0.1) | 3 (0.1) | - |
| Multiracial | 64 (1.2) | 49 (1.2) | 26 (1.4) |
| Other Race NH | 31 (0.6) | 23 (0.6) | 10 (0.5) |
| Don't Know/Refused | 143 (2.7) | 84 (2.1) | 52 (2.8) |
| **Education** | | | |
| <High School | 531 (10.2) | 369 (9.3) | 226 (12.2) |
| High School | 1354 (26.0) | 1025 (26.0) | 516 (27.8) |
| Some Post High School | 1317 (25.3) | 985 (25.0) | 467 (25.1) |
| College graduate | 1984 (38.1) | 1555 (39.4) | 646 (34.7) |
| Don't Know/Refused | 25 (0.5) | 13 (0.3) | 4 (0.2) |
| **Annual Income (USD)** | | | |
| <$25,000 | 1250 (24.0) | 945 (23.9) | 552 (29.7) |
| $25,000–$49,999 | 1020 (19.6) | 746 (18.9) | 342 (18.4) |
| $50,000+ | 1934 (37.1) | 1628 (41.2) | 683 (36.7) |
| Don't Know/Refused | 1007(19.3) | 628 (15.9) | 282 (15.2) |
| **Marital Status** | | | |
| Couple | 2657 (51.0) | 2167 (54.9) | 934 (50.2) |
| Single | 2498 (47.9) | 1756 (44.5) | 915 (49.2) |
| Don't Know/Refused | 56 (1.1) | 24 (0.6) | 10 (0.5) |
| **Healthcare Coverage** | | | |
| Yes | 4780 (91.7) | 3567 (90.4) | 1547 (83.2) |
| No | 409 (7.8) | 365 (9.2) | 305 (16.4) |
| Don't Know/Refused | 22 (0.5) | 15 (0.4) | 7 (0.4) |

**Table 3.** Prevalence of colonoscopy and sigmoidoscopy utilization for colorectal cancer screening in Georgia: BRFSS 2018 Data.

| Variable | Respondents Who Have Ever Had Sigmoidoscopy or Colonoscopy | | Respondents Who Have Had Colonoscopy in the Past 10 Years | | Respondents Who Have Had Sigmoidoscopy in the Past 5 Years | |
|---|---|---|---|---|---|---|
| | Unweighted N | Weighted % | Unweighted N | Weighted % | Unweighted N | Weighted % |
| Overall | 3711 | 68.3 | 2580 | 62.9 | 69 | 3.5 |
| **Sex** | | | | | | |
| Male | 1473 | 66.9 | 1056 | 61.2 | 27 | 3.0 |
| Female | 2236 | 69.5 | 1523 | 64.5 | 42 | 3.9 |
| **Age (years)** | | | | | | |
| 50–59 | 867 | 57.7 | 774 | 53.3 | 24 | 2.4 |
| 60–69 | 1326 | 73.7 | 1168 | 67.6 | 26 | 3.1 |
| 70–79 | 1046 | 80.7 | 638 | 77.5 | 19 | 8.8 |
| 80+ | 398 | 72.3 | - | - | - | - |
| **Race** | | | | | | |
| White NH | 2589 | 71.8 | 1745 | 64.8 | 36 | 2.8 |
| Black NH | 836 | 65.9 | 636 | 63.3 | 22 | 4.8 |
| Hispanic | 80 | 40.8 | 64 | 38.4 | 3 | 3.3 |
| American Indian/ Alaskan Native | 32 | 65.4 | 24 | 66.4 | 0 | 0.0 |
| Asian | 23 | 64.4 | 20 | 69.4 | 0 | 0.0 |
| Native Hawaiian/ Pacific Islander | 4 | 88.1 | 3 | 100.0 | - | - |
| Multiracial | 41 | 62.6 | 29 | 57.4 | 3 | 12.4 |
| Other Race NH | 19 | 55.5 | 13 | 53.0 | 0 | 0.0 |
| **Education** | | | | | | |
| <HS | 294 | 55.6 | 175 | 50.5 | 12 | 5.0 |
| High School | 893 | 64.5 | 612 | 57.9 | 11 | 2.1 |
| Some PHS | 948 | 71.7 | 637 | 64.8 | 23 | 4.4 |
| College grad | 1559 | 75.8 | 1146 | 72.1 | 23 | 3.2 |
| **Annual Income (USD)** | | | | | | |
| <$25,000 | 767 | 58.4 | 494 | 51.3 | 23 | 4.1 |
| $25,000–$49,999 | 741 | 69.0 | 493 | 62.0 | 13 | 2.7 |
| $50,000+ | 1517 | 75.5 | 1187 | 70.3 | 24 | 3.1 |
| **Marital Status** | | | | | | |
| Couple | 1988 | 71.5 | 1525 | 67.1 | 33 | 3.3 |
| Single | 1684 | 63.9 | 1039 | 56.4 | 35 | 3.8 |
| **Healthcare Coverage** | | | | | | |
| Yes | 3568 | 72.4 | 2474 | 67.4 | 62 | 3.9 |
| No | 131 | 30.1 | 97 | 25.8 | 7 | 1.6 |

*3.4. Adjusted Odds of Colonoscopy and Sigmoidoscopy Utilization for Colorectal Cancer Screening*

Sociodemographic covariates of sex, age, race, education, income, marital status, and healthcare coverage were adjusted for, and the results of the adjusted model from logistic regression are displayed in Table 4.

**Table 4.** Adjusted odds of utilizing colonoscopy or sigmoidoscopy for colorectal cancer screening in Georgia: BRFSS 2018 Data.

| Variable | Ref | Colonoscopy or Sigmoidoscopy | | Colonoscopy | | Sigmoidoscopy | |
|---|---|---|---|---|---|---|---|
| | | Odds Ratio (95% C.I) | *p*-Value | Odds Ratio (95% C.I.) | *p*-Value | Odds Ratio (95% C.I.) | *p*-Value |
| **Sex** | | | | | | | |
| Female | Male | 1.20 (1.05, 1.38) | 0.008 | 1.21 (1.04, 1.39) | 0.012 | 1.17 (0.70, 1.95) | 0.54 |
| **Age** (years) | | | | | | | |
| 60–69 | 50–59 | 2.38 (2.02, 2.80) | <0.001 | 2.09 (1.78, 2.45) | <0.001 | 1.16 (0.65, 2.08) | 0.61 |
| 70–79 | | 2.88 (2.38, 3.48) | <0.001 | 2.52 (2.05, 3.08) | <0.001 | 2.12 (1.10, 4.12) | 0.26 |
| **Race** | | | | | | | |
| Black | White | 1.04 (0.89, 1.22) | 0.64 | 1.30 (1.10, 1.55) | 0.003 | 2.06 (1.17, 3.64) | 0.013 |
| Hispanic | | 0.68 (0.46, 1.02) | 0.06 | 0.64 (0.40, 1.03) | 0.067 | - | - |
| **Education** | | | | | | | |
| High School | <High School | 1.32 (1.05, 1.65) | 0.017 | 1.32 (1.01, 1.71) | 0.040 | 0.38 (0.16, 0.89) | 0.025 |
| Some PHS | | 1.63 (1.29, 2.06) | <0.001 | 1.49 (1.14, 1.95) | 0.004 | 0.93 (0.43, 2.00) | 0.850 |
| College grad | | 2.08 (1.64, 2.63) | <0.001 | 1.98 (1.51, 2.59) | < 0.001 | 0.72 (0.31, 1.63) | 0.424 |
| **Annual Income** (USD) | | | | | | | |
| $25,000–$49,999 | <$25,000 | 1.24 (1.01, 1.51) | 0.037 | 1.33 (1.07, 1.65) | 0.011 | 0.86 (0.42, 1.79) | 0.693 |
| $50,000+ | | 1.56 (1.27, 1.91) | <0.001 | 1.60 (1.28, 1.99) | <0.001 | 0.85 (0.41, 1.77) | 0.662 |
| **Marital Status** | | | | | | | |
| Couple | Single | 1.20 (1.04, 1.39) | 0.012 | 1.38 (1.18, 1.61) | <0.01 | 1.11 (0.64, 1.90) | 0.71 |
| **Healthcare Coverage** | | | | | | | |
| Yes | No | 3.86 (3.05, 4.88) | <0.001 | 3.88 (2.99, 5.03) | <0.001 | 1.71 (0.74, 3.96) | 0.21 |

The odds of colonoscopy utilization were higher for respondents who were female (vs. male [aOR = 1.21, C.I. = 1.04, 1.39]), older (60–69 vs. 50–59 [aOR = 2.09, C.I. = 1.78, 2.45], 70–79 vs. 50–59 [aOR = 2.52, C.I. = 2.05, 3.08]), Black (vs. White [aOR = 1.30, C.I.= 1.10, 1.55]), and in a couple relationship (vs. single [aOR = 1.38, C.I. = 1.18, 1.61]). The odds of colonoscopy utilization were also higher for those with higher education attainment (high school vs. <high school [aOR = 1.32, C.I.= 1.01, 1.71], some post high school vs. <high school [aOR = 1.49, C.I. = 1.14, 1.95], college graduate vs. <high school [aOR = 1.98, C.I. = 1.51, 2.59]) earning a higher income ($25,000–$49,999 vs. <$25,000 [aOR = 1.33, C.I. = 1.07, 1.65], $50,000+ vs. <$25,000 [aOR = 1.60, C.I. = 1.28, 1.99]), and having healthcare coverage (yes vs. no [aOR = 3.88, C.I. = 2.99, 5.03]).

The odds of sigmoidoscopy utilization were significantly higher for respondents who were Black (vs. White [aOR = 2.06, C.I. = 1.17, 3.64]) and those with a less than high school education (vs. high school [aOR = 0.38, C.I. = 0.16, 0.89]).

## 4. Discussion

The findings from this study show an increase over time in the utilization of colonoscopy or sigmoidoscopy for CRC screening among adults 50 years and older in Georgia. The 2018 BRFSS data revealed differences between sociodemographic groups, with individuals who were more likely to utilize colonoscopy being female, older, Black, and in a couple relationship, with higher education attainment, higher income, and healthcare coverage. The likelihood of sigmoidoscopy use was higher among Black individuals and those with a less than high school education.

In support of the current findings, previously published studies reported rising trends in colonoscopy utilization over time [36–38]. Shapiro et. al. [36] observed an overall increase in the use of colonoscopy among adults aged 50 to 75 in the USA from 57% in 2010 to 61% in 2018. Colonoscopy use was significantly lower for adults aged 50–64 years who were never married and were uninsured. Colonoscopy use was observed to be lower among non-Hispanic Blacks in contrast to the findings of the present study. Lieberman et. al. [37] reported a threefold increase in screening colonoscopy from 2000 to 2011 among adults included in the National Endoscopic Database. Richards et. al. [38] observed an increase from 41.7% in 2003 to 61.7% in 2007 among adults in the state of New York. Results from the study by May et. al. [39] showed that between 2008 and 2016, colonoscopy was the most used screening modality, with utilization rates rising from 74.9% to 83.7%, while sigmoidoscopy use decreased from 2.9% to 0.7%. The observed increase over time noted in the present study may be due mainly to the steep rise in colonoscopy use, because the 2018 data shows the overall weighted prevalence of colonoscopy use was 62.9% and that of sigmoidoscopy use was 3.5% (Table 3).

The rise in colonoscopy use over time has been attributed to several factors. Physician preference for and recommendation of colonoscopy may be a major factor for the rising colonoscopy rates. Some studies have shown that physicians often consider colonoscopy to be the gold standard for CRC screening [19,40,41]. The entire colon can be examined by colonoscopy, and it allows for the removal of precancerous polyps during the procedure. It only needs to be performed every 10 years, unlike the other screening modalities which are repeated more frequently [42]. Another contributing factor to the rise in colonoscopy use is the coverage by the Medicare program for average risk individuals which began in 2001 [42], and the implementation of the Affordable Health Care Act which provides coverage for CRC screening without co-payments [43]. These factors may have led to the increased ordering of colonoscopy by physicians, as evidenced by previous studies that found a dramatic increase in the use of colonoscopy procedures after Medicare coverage was enacted [42,44,45].

The observed differences in colonoscopy utilization trends between Georgia and the nation (AAPC: 2.3 for Georgia vs. 2.9 nationally) may be attributed to the fact that there are more individuals without healthcare coverage in Georgia when compared with nationwide figures. In 2021, 12.7% of individuals living in Georgia compared to 8.6% nationally were without healthcare coverage [46]. This has led to the higher utilization of the cheaper blood stool tests among Georgia residents [25].

Despite the rise in colonoscopy utilization, screening rates for CRC have remained lower than the national goals of 70.5% and 74.4% set by Healthy People 2020 and 2030, respectively. This may be due to the declining utilization of other screening modalities such as blood stool tests [25] and sigmoidoscopy [39]. The attainment of the nationally set goals for CRC screening may not be achievable, because the current capacity may be insufficient to provide a colonoscopy to most eligible adults who have not been screened [43]. Colonoscopy is the most invasive and costly screening modality, and can only be performed by trained specialists. The number of providers who are qualified to perform colonoscopies is relatively small; therefore, accommodating the significant increases in demand for such services may be difficult [42]. Blood stool test as a first line of screening can drastically reduce the costs of screening infrastructure, especially for populations with relatively low risks of CRC. Programs that are based on FIT can increase the yield of colonoscopy, such

that 1 CRC is found in approximately every 11 to 33 follow-up colonoscopy procedures, compared with 1 in approximately 200 screening colonoscopy procedures without an initial FIT [47,48].

The nationally set goals for CRC screening may likely be achieved if individuals can make informed choices about their preferred screening methods [42,49]. A previous study [50] reported variations among patients' preferences for CRC screening modalities. About 37% of patients preferred colonoscopy, compared to 31% and 9% who preferred blood stool test and sigmoidoscopy, respectively. Further educational interventions are needed to provide clinicians with complete understanding of the CRC screening process, including up-to-date guidelines for recommended screening modalities and consideration of patient, clinician, and health system factors that may impact the effectiveness of each method [9].

*Study Limitations*

The BRFSS surveys for the earlier years that were included in this present trend analysis did not have separate questions for colonoscopy and sigmoidoscopy. The question asked was "Have you ever had a colonoscopy or sigmoidoscopy?" The observed increase in utilization was mainly due to the increase in colonoscopy use. In 2018, the overall prevalence of colonoscopy use was much higher than that of sigmoidoscopy, and a similar study reported a decline in the use of sigmoidoscopy over time for CRC screening [39]. Self-report is used for BRFSS surveys, thus recall bias is another limitation that may result in overestimation, underestimation, or misclassification of the presented findings. Despite these limitations, data from the BRFSS are reliable and generally valid.

## 5. Conclusions

There is a steady rise in colonoscopy utilization for CRC screening in Georgia that is less pronounced when compared nationally. This rise is associated with several factors that include socioeconomics. The current CRC screening rates remain below the national target despite the rise in colonoscopy use. Educational interventions that promote informed screening recommendations among clinicians, consider patients' preferences, and address socioeconomic disparities are needed for achieving the desired CRC screening rates. Future research that compares the utilization rates of CRC screening methods between national and international regions may reveal the correlation between screening methods, screening adherence, and CRC rates. In addition, the current guidelines that lower the eligible age for CRC screening from 50 to 45 may change future utilization patterns of CRC screening methods.

**Author Contributions:** Conceptualization, B.E.A., methodology, B.E.A., Z.H., and N.L.; software, B.E.A., Z.H., and N.L.; validation, B.E.A.; formal analysis, B.E.A., Z.H., and N.L.; investigation, B.E.A., Z.H., and N.L.; resources, B.E.A., Z.H., N.L., and J.A.J.; data curation, B.E.A., Z.H., and N.L.; writing—original draft preparation, B.E.A. and Z.H.; writing—review and editing, B.E.A., Z.H., N.L., B.D., K.M.I., and J.A.J.; supervision, B.E.A. All authors have read and agreed to the published version of the manuscript.

**Funding:** This research received no external funding.

**Institutional Review Board Statement:** Publicly available BRFSS datasets have no individually identifiable information and do not meet the requirements of human subjects research as defined by the U.S. Department of Health and Human Services. Therefore, additional IRB approval is not required for the use of these datasets.

**Informed Consent Statement:** The current study was a secondary analysis of the BRFSS datasets and did not require informed consent from participants. The collection and release activities of BRFSS datasets are governed by rules, regulations, and legislative authorizations.

**Data Availability Statement:** Data for this study is publicly available at https://www.cdc.gov/brfss/annual_data/annual_data.htm (accessed on 1 April 2021).

**Acknowledgments:** The authors thank the faculty and staff of the Institute of Public and Preventive Health, Augusta University, for the opportunity afforded to Zachary Hoffman and Nicollette Lewis to participate in the Summer Public Health Scholars Program and be part of this project.

**Conflicts of Interest:** The authors declare no conflict of interest.

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
