# Peer review of "Evaluation of Colonoscopy and Sigmoidoscopy Utilization for Colorectal Cancer Screening in Georgia, USA"

_curroncol, doi:10.3390/curroncol29110703_

Round 1

Reviewer 1 Report

Thank your very much for offering me the privilege of reviewing the paper entitled "Evaluation of Colonoscopy and Sigmoidoscopy utilization for 2 Colorectal Cancer Screening in Georgia, USA" by Ansa BE et al. 

This is an elaborate and extensive sociodemographic study, upon which, being written by native American-English speaking specialist authors, not much can be commented upon.

The only major comment to the study (which has already been addressed by the authors in the “potential biases” section) is the one related to the bias offered by the non-stratification of answer and patients’ subpopulation when the subjects were asked if they undertook a “colonoscopy or sigmoidoscopy”. Perhaps further “table-based” socio-demographic descriptive statistics regarding subgrouping these patients (if possible), could add additional strength to the study.

The second concern regarding publishing of this relevant, but US-based regional study is the fact that it cannot be extrapolated to international data, except the national US NIH published data. Perhaps, the authors could (if they wish so), at least, insert a paragraph comparing their findings to international socio-demographic data, liberally available.

Author Response

Comment: Thank you very much for offering me the privilege of reviewing the paper entitled "Evaluation of Colonoscopy and Sigmoidoscopy utilization for Colorectal Cancer Screening in Georgia, USA" by Ansa BE et al.

This is an elaborate and extensive sociodemographic study, upon which, being written by native American-English speaking specialist authors, not much can be commented upon.

Response: We appreciate your kind comment and thank you also for giving the time to review our manuscript. Your valuable comments have helped to improve our paper.

Comment: The only major comment to the study (which has already been addressed by the authors in the “potential biases” section) is the one related to the bias offered by the non-stratification of answer and patients’ subpopulation when the subjects were asked if they undertook a “colonoscopy or sigmoidoscopy”. Perhaps further “table-based” socio-demographic descriptive statistics regarding subgrouping these patients (if possible), could add additional strength to the study.

Response: Thank you for mentioning this important issue. We also observed this limitation and mentioned it in the paper. It is difficult to subdivide the subjects for previous years because the data collected did not address this. That is the reason why we subdivided the subjects using the 2018 dataset that had the necessary information (see tables 2 and 3). We hope that this will add additional strength to the study.

Comment: The second concern regarding publishing of this relevant, but US-based regional study is the fact that it cannot be extrapolated to international data, except the national US NIH published data. Perhaps, the authors could (if they wish so), at least, insert a paragraph comparing their findings to international socio-demographic data, liberally available.

Response: We thank you again for this important suggestion. The main objective of the current study was focused on Georgia, USA. We plan to compare these findings with those from other national and international regions in a later study. We added a sentence in the conclusion to reflect this: “ Future research that compares the utilization rates of CRC screening methods between national and international regions may reveal the correlation between screening methods, screening adherence, and CRC rates.”

Reviewer 2 Report

This is a well written article on demographic factor that are related to using endoscopic means of CRC screening. It is well written and has sufficient references.

Introduction  I would add the know mortality rate reduction with using colonoscopy, Flex. Sig. and FIT testing since it's been studied. The S and S are useful but this info more germane.

M+M

The BRFSS rate of 43.6 is low and one wonders those that don't have phones might make the numbers even off more. The other point here is you asked in 2018 data if you ever got a Colonoscopy in the past 10 years or Flex Sig in the past 5 years.  But that doesn't mean compliance of a prevention program.  For example, If you had one once had polyps and not following repeated screening due already you are out of compliance but, if you are not due to get one this year you are in compliance. Please clarify why this isn't ideal. Also, the trend for the past few years of colonoscopy over flex. sig is wide spread. But in Georgia what might be the reason which seems Less?

Discussion

With the new guidelines of screening at age 45 any sense of the direction your data may change. Might comment since older age increases compliance. 

Author Response

Comment: This is a well written article on demographic factors that are related to using endoscopic means of CRC screening. It is well written and has sufficient references.

Response: Thank you for your kind comment and we appreciate the time that you have spent in reviewing our manuscript.

Introduction

Comment: I would add the known mortality rate reduction with using colonoscopy, Flex. Sig. and FIT testing since it's been studied. The S and S are useful but this info more germane.

Response: Thank you for this important suggestion. We have added a sentence in the introduction to this effect: “The FIT and fecal occult blood test (FOBT) reduce mortality from CRC by 40% and 15%–33% respectively, compared to 13%–50% for sigmoidoscopy, and 60%–75% for colonoscopy [12–15].”

M+M

Comment: The BRFSS rate of 43.6 is low and one wonders those that don't have phones might make the numbers even off more.

Response: You are correct that those without phones (non-coverage) may make the overall estimates more than reported. This was adjusted for in the current study by using weighted percentages. See statistical analysis: “Crosstabs were done to calculate weighted percentages of respondents who reported having had colonoscopy or sigmoidoscopy in 2018. Data were weighted to adjust for non-coverage, non-response, and for generalization of results [33].”

Comment: The other point here is you asked in 2018 data if you ever got a Colonoscopy in the past 10 years or Flex Sig in the past 5 years.  But that doesn't mean compliance of a prevention program.  For example, if you had one once had polyps and not following repeated screening due already you are out of compliance but, if you are not due to get one this year you are in compliance. Please clarify why this isn't ideal.

Response: You are correct with this observation. In the current study, we focused on the utilization rates of colonoscopy and sigmoidoscopy in Georgia, and not compliance. We have another study in progress that focused on compliance/adherence rates and addresses this point. We hope to submit our findings soon for publication.

Comment: Also, the trend for the past few years of colonoscopy over flex. sig is widespread. But in Georgia what might be the reason which seems Less?

Response: We have added a sentence in the discussion to explain the possible reason: “The observed differences in colonoscopy utilization trends between Georgia and nationally (AAPC: 2.3 for Georgia versus 2.9 nationally) may be attributed to the fact that there are more individuals without healthcare coverage in Georgia compared to nationally. In 2021, 12.7% of individuals living in Georgia were without healthcare coverage compared to 8.6% nationally [47]. This has led to the higher utilization of the cheaper blood stool tests among Georgia residents [25].”

Discussion

Comment: With the new guidelines of screening at age 45 any sense of the direction your data may change. Might comment since older age increases compliance.

Response: We have included a sentence in the conclusion to address this suggestion: “…the current guidelines that lower the eligible age for CRC screening from 50 to 45 may change future utilization patterns of CRC screening methods.”